# Silage Fermentation: A Potential Biological Approach for the Long-Term Preservation and Recycling of Polyphenols and Terpenes in Globe Artichoke (*Cynara scolymus* L.) By-Products

**DOI:** 10.3390/molecules25143302

**Published:** 2020-07-21

**Authors:** Zhuoyan Fan, Kai Chen, Lingyin Ban, Yu Mao, Caiyun Hou, Jingming Li

**Affiliations:** College of Food Science and Nutritional Engineering, China Agricultural University, Beijing 100083, China; FZY2830@163.com (Z.F.); kai.chen@cau.edu.cn (K.C.); banlingyin@hotmail.com (L.B.); m1758782709@163.com (Y.M.)

**Keywords:** globe artichoke by-product, silage, polyphenol, terpene, lactic acid bacteria

## Abstract

An economic and effective method for storage is necessary to make full use of the nature of active components in artichoke by-products and ease environmental pressure. In this paper, the potential of silage fermentation for the preservation and recycling of polyphenols and terpenes in artichoke by-products is evaluated. The silage of artichoke by-products is characterized by lactic acid bacteria fermentation. Silage distinctly increases the abundance of lactic acid bacteria in artichoke by-products, such as *Lactobacillus*, *Lactococcus*, *Serratia*, and *Weissella*, and greatly increases the abundance of *Firmicutes*. The improvement of the microorgan structure and composition is of great significance for the quality of artichoke by-products. Polyphenols in the stems and leaves of artichokes are preserved well in silage. Among the 18 polyphenol compounds detected by high performance liquid chromatography coupled with triple quadrupole tandem mass spectrometry (HPLC-QqQ-MS/MS), the contents of 11 phenolic acids and four flavonoids increased significantly. For terpenes detected by gas chromatography-mass spectrometry (GC-MS), the contents of four pentacyclic triterpenoids increased significantly, while two sterols were kept stable in the silage process. Silage is a potential biotechnology for the long-term preservation of bioactive components, such as polyphenols and terpenes in artichoke by-products, and the results provide a scientific basis for the efficient utilization of by-products.

## 1. Introduction

As the public are becoming increasingly health-conscious, many natural phytochemicals with extensive antioxidant and anti-inflammatory activities have drawn attention [1]. Artichokes (*Cynara scolymus L*.), a perennial herb belonging to the Asteraceae family, are widely cultivated in Brazil, Italy, France, the southern United States, South America, and China. The edible part of the artichoke is the immature inflorescence, which is rich in bioactive compounds [2,3,4]. On the other hand, the stems and leaves of the artichoke also contain various active compounds (polyphenols, terpenes, sterols, etc.) [4,5,6,7]. However, the by-products of the plant, which have a large biomass (about 80–85% of the plant), are the non-edible parts that are mostly abandoned because of no appropriate storage method, except for small amounts which are used to raise animals [8,9].

Megías et al. believe that silage is the most promising method for the long-term preservation of artichoke by-products [10]. Silage is a crop preservation method based on natural lactic acid fermentation under anaerobic conditions [11]. Silage can be used as an economical, low-consumption, and environmentally friendly preservation method, meeting the annual demand for high-quality raw materials for industrial production and increasing the value of artichoke by-products. Many researchers have focused on the changes of carbohydrates, proteins, and other nutrients in artichoke by-products, but fewer have focused on the functional components [10,11,12,13]. We focus on the secondary metabolites in artichoke by-products while in silage here, such as polyphenols and terpenes, which possess antioxidant and anti-inflammatory activities. We aim to provide a scientific basis for the efficient utilization of artichoke by-products for developing effective anti-aging supplements.

## 2. Results

### 2.1. Effects of Silage on Microorganisms in Artichoke By-Products

Silage is a process involving complex microbial activities and nutrient changes. This process improves fermentation quality by promoting changes in the microbial composition [14]. The dominant microorganisms directly affect the quality of silage, and lactic acid bacteria are the main fermentation bacteria [15,16,17,18]. In order to monitor the fermentation process of artichoke by-products, we analyzed the dynamic changes in the microbial community structure (Figure 1 and Figure 2), and we used a species accumulation curve to investigate the scientific value of the sampling (Appendix A).

Our results show that the dominant microorganism at the phylum level before silage is *Proteobacteria* (99.24%). Although *Proteobacteria* is always the dominant microorganism during silage, silage inhibits the proportion of it (83.98%) and greatly increases the abundance of *Firmicutes* (from 0.36% to 21.08%) (Figure 1). At the same time, the dominant microbiota in the raw materials before silage were *Pseudomonas* and *Acinetobacter;* they came to be *Lactobacillus*, *Lactococcus*, *Leuconostoc*, *Serratia*, *Weissella*, and *Enterococcus* after silage. The results show that silage distinctly increased the abundance of lactic acid bacteria of artichoke by-products and effectively decreased the abundance of harmful bacteria, such as *Pseudomonas* (from 29.35% to 0.05%) and *Acinetobacter* (from 2.44% to 0.05%) (Figure 2). In addition, we analyzed the abundance information (the top 35 genera) of each sample at the genus and sample levels. In later silage, the growth of aerobic bacteria was inhibited, which created conditions for the growth and reproduction of *Lactobacillus* in previous silage processes (Figure 3). Our results suggest that the silage with artichoke by-products is characterized by lactic acid bacteria fermentation. The improvement of the microorgan structure and composition is of great significance for the quality of artichoke by-products.

### 2.2. Effects of Silage on Polyphenols in Artichoke By-Products

Silage effectively preserves the chemical components in artichoke by-products [12,13]. In order to explore the effect of silage on the polyphenols in artichoke by-products, 18 polyphenols were qualitatively and quantitatively analyzed by HPLC-QQQ-MS/MS. The results show that polyphenols in the stems and leaves of the artichoke (Table 1 and Table 2) are preserved well in silage. Among the 18 polyphenol compounds detected, the contents of 11 small molecular weight phenolic acids (top 11 compounds in Table 1) and four flavonoids (top 4 compounds in Table 2) increased significantly (*p* < 0.05). Although the 4-*O*-caffeoylquinic acid and phlorizin contents were not markedly increased (*p* > 0.05), they were well-preserved in silage. In addition, we found the contents of apigenin-7-*O*-glucuronide, protocatechuic acid, gallic acid, and (−)-epicatechin gallate reached higher levels in the early silage (i.e., on day 2 or 5, *p* < 0.05), suggesting that high levels of different polyphenols could be obtained by regulating the silage time. 

### 2.3. Effects of Silage on Terpenoids in Artichoke By-Products

In addition to polyphenols, artichoke by-products are also abundant in terpenoids [7,19,20], which have strong antioxidant, anti-inflammatory, anti-cancer, and other physiological activities. Rich experience has been accumulated in recent studies about the structures and activities of terpenoids [21,22], and we have carried out qualitative and quantitative analysis of pentacyclic triterpenoids and sterols. We detected four pentacyclic triterpenoids (α-amyrin, lupeol, Ψ-taraxasterol, and taraxasterol) and two sterols (stigmasterol and β-sitosterol) in fat-soluble extracts of artichoke by-products. Interestingly, as seen in the Figure 4, we found that the contents of all pentacyclic triterpenoids increased significantly on the second day of silage (contents of four pentacyclic triterpenoids increased from 206.20, 423.06, 249.35, and 606.18 mg/kg DM to 589.40, 1762.35, 1188.63, and 1187.96 mg/kg DM, respectively) (*p* < 0.05), and then remained stable after the fifth day of silage, and this was maintained at high levels at the end of silage (contents were 440.93, 1494.39, 993.61, and 853.94 mg/kg DM, respectively). For sterols, the contents of stigmasterol and β-sitosterol had no significant change in the entire silage process (*p* > 0.05) (Figure 5). Nevertheless, our results suggested that silage storage results in the terpenes in artichoke by-products being well-preserved, which is a method especially suitable for long-term preservation.

### 2.4. Correlation Analysis between the Structure of Bacterial Community and Contents of Polyphenols and Terpenes

In order to further clarify the effect of the microbial community structure on the polyphenols and terpenes of artichoke by-products during the silage process, we carried out correlation analysis for the relative abundance of microbiota and the content change of polyphenols and terpenes. The results show that (Figure 6) in the process of silage fermentation, the contents of polyphenols and terpenes are significantly associated with the relative abundance of microbiota in the artichoke by-products. Particularly, the abundance of *Pseudomonas* and *Acinetobacter* was negatively correlated with the contents of polyphenols and terpenoids here (*p* < 0.01). Meanwhile, there was a positive correlation between the abundance of *Lactobacillus*, *Enterococcus*, *Serratia*, and *Lactococcus* and the contents of polyphenols and terpenes here (*p* < 0.01). The plant secondary metabolites resisted the invasion of *Pseudomonas* and *Acinetobacter*, promoting the growth of lactic acid bacteria. In turn, the lactic acid bacteria promoted the production and accumulation of polyphenols and terpenes and inhibited the growth of harmful bacteria (proposed model depicted in Figure 7).

## 3. Discussion

Polyphenols and terpenoids are secondary metabolites of plants, and both play important roles in plant growth, environmental adaptation, and resistance to diseases and pests. In addition to their important biological and ecological significance, these secondary metabolites are also widely utilized as medicine with strong efficacy, especially for anti-inflammatory and antioxidant uses [23,24,25]. Artichoke by-products, such as bracts, stems, and leaves, are rich in caffeoylquinic acids, flavonoids, and triterpenoids [7,26,27]. However, these by-products are often piled up and discarded in processing, which not only causes a waste of material resources, but also increases pressure on the environment. Finding an economic and reasonable storage method is necessary for making full use of the functional components of artichoke by-products, because these by-products have high water contents and strong seasonal effects that are liable to suffer spoilage [9,12]. In addition, a high water content will increase the drying cost. Silage is commonly used as a crop preservation method based on natural lactic acid fermentation under anaerobic conditions, which can improve the quality of crops [11]. In this study, the dynamic changes of polyphenols and terpenes in artichoke by-products in silage were investigated. We evaluated the potential of silage fermentation for the preservation and recycling of the active components in artichoke by-products.

A good fermentation process in silage is the necessary condition to ensure the quality of artichoke by-products [24]. We firstly analyzed the variations of the microbial communities of artichoke by-products in the process of silage fermentation. The compositions and abundances of bacteria were constantly changing, and a few predominant bacteria (such as *Lactobacillus*) played determinative roles [28]. The two phyla that dominated after silage were *Proteobacteria* and *Firmicutes* (Figure 2), which is similar to what was found in previous studies [29,30]. The increased abundance of *Firmicutes* might be related to the increased abundance of lactic acid bacteria such as *Lactobacillus*, *Lactococcus*, *Enterococcus*, and *Leuconostoc* (Figure 1). The trend of the microbial community structure was beneficial to good nutritional and hygienic quality of the artichoke by-products.

Studies on the active components and functions of artichokes have mostly focused on polyphenols [27,28,30,31,32,33]. During the growth of artichokes, polyphenols tend to gather on the outer parts (outer bracts, stems, leaves, etc.) in order to better exert biological functions [5]. On the other hand, the concentration of polyphenols is affected by many factors, such as processing, storage, and agronomic management [34,35,36,37]. We found that the polyphenols in the by-products of silage artichokes were well-preserved after good lactic acid fermentation. A total of 18 polyphenols were detected, among which 17 were well-preserved during silage (including 12 small molecular phenolic acids and five flavonoids) (Table 1 and Table 2). However, the content of chlorogenic acid decreased, which might the degrade in the presence of polyphenol oxidase and lactic acid [38]. The results were similar to those in the previous study done by Garbetta et al., who used *Lactobacillus paraccasei* (LMG P-22043) as the fermentation strain and found that the 1-*O*-caffeoylquinic acid content in artichokes increased while the contents of chlorogenic acid and 3-*O*-caffeoylquinic acid decreased [39]. Anyhow, our results suggest that the polyphenols in the artichoke by-products were well-preserved after the silage process, especially for small-molecule phenolic acids.

Terpenoids are one of the three major secondary metabolites, among which pentacyclic triterpenoids are common in medicinal plants. Terpenoids containing a pentacyclic triterpenoid mother nucleus structure possess strong anti-inflammatory and antioxidant activities [40,41,42,43]. We detected two pentacyclic terpenoids in the artichoke by-products, including α-amyrin, lupeol, Ψ-taraxasterol, and taraxasterol. The drastic changes of different substances mainly occurred in the early stage of silage (the day 2 of silage). That might be due to the plant respiration that was ongoing, and the enzymes and aerobic microorganisms that still had certain activities at the early stage of silage. Besides, anaerobic microorganisms gradually increased, which made the silage environment system unstable and increased the production of secondary metabolites. Phytosterols, belonging to triterpenoids, are important components for plant cells. The most common phytosterols are stigmasterol, sitosterol, and brassinosteroids [44,45]. Unlike pentacyclic triterpenoids, the contents of stigmasterol and β-sitosterol were stable throughout the silage process and were well-preserved.

## 4. Materials and Methods

### 4.1. Sample Preparation

Three independent biological replications of artichoke (*Cynara scolymus L.*) stems and leaves that grew well were collected with an approximately homogenous size. The samples were hand-harvested by the same person at the same maturity from Huaqiao Farm (103.705089° E, 25.024768° N) in Zhongshu Town, Luliang County, Qujing City, Yunnan Province, China. After harvesting, they were collected in thermocol boxes and delivered to the laboratory as soon as possible. The stems and leaves of the artichokes were cut into 3–5 cm segments. After fully mixing, the small segments were randomly and evenly packed in 30 polyethylene packaging bags (25 cm × 25 cm) under vacuum and stored in 23 ± 2 °C for silage, and each bag weighed almost 200 g. Three bags were randomly taken on days 0, 2, 5, 10, 20, 30, and 60 of silage for independent analysis, respectively.

Except for samples for microbiota change analysis, the artichoke by-product samples used in this study were freeze-dried. After crushing, the samples were sieved through a 50 mesh and then refrigerated at −20 °C. Reagents and solvents were analytical or HPLC grade. Deionized water was used throughout. The standard samples used for detection were purchased from Sigma (Sigma-Aldrich, 3050 Spruce St. St Louis, MO, USA).

### 4.2. Analysis of Microbiota Change

DNA extraction was carried out as described previously by Keshri et al. [29]. Here, 10 g of artichoke by-products (on day 0, 2, 5, 10, 20, 30 and 60 of silage, respectively) were taken and 40 mL of sterile water was added. After being shaken at 120 rpm for 2 h and centrifugated at 12,800× *g* for 30 min, the sediment was stored at −80 °C. The total genomic DNA was extracted by the method of sixteen alkyl three ethyl ammonium bromide (CTAB) and twelve alkyl sulfonate (SDS). The DNA concentration and purity were detected by agarose gel electrophoresis and the DNA was diluted to 1 ng/μL via sterile water.

Quantitative PCR detection: (1) PCR amplification: Diluted genomic DNA was used as a template. Specific primers with Barcode 515F-806 R were used to amplify 16S rDNA genes in the 16S V4 region. (2) PCR reaction system: Phusion^®^ Super-Fidelity PCR Master Mix 15 μL, 0.2 μmol/L of upstream and downstream primers, and 10 ng of DNA template. The volume was made up to 30 μL with deionized water. (3) PCR reaction program: 98 °C pre-denaturation for 1 min; 30 cycles of 98 °C, 10 s; 50 °C, 30 s; 72 °C, 30 s; then 72 °C for 5 min.

Mixing and purification of PCR products: PCR products were electrophoresed using a 2% agarose gel. Samples were mixed at equal concentrations according to the PCR product concentration. After thorough mixing, PCR products were detected using 2% agarose gel electrophoresis. Products were recycled by a Thermo Scientific Gene JET Gel Recovery Kit (Thermo Scientific, Waltham, MA, USA).

Library construction and sequencing: The library was constructed using the NEB Next^®^ Ultra TM DNA Library Prep Kit (New England Biolabs, Ipswich, MA, USA) for Illumina from New England Biolabs. The constructed library was subjected to Qubit quantification and library detection, and Mi Seq high-throughput sequencing technology was used for on-line sequencing.

### 4.3. Analysis of Polyphenols

Polyphenols were extracted as described previously by Rouphael et al. and Jin et al. [46,47]. Here, 1 g of powder of freeze-dried artichoke by-product (on day 0, 2, 5, 10, 20, 30, and 60 of silage) was extracted with 10 mL of 70% methanol. After being ultrasonically extracted in an ice water bath for 30 min and centrifugated at 4000× *g* (4 °C) for 20 min, the supernatant was collected.

Polyphenols were detected by high performance liquid chromatography coupled with triple quadrupole tandem mass spectrometry (HPLC-QqQ-MS/MS): The polyphenol extracts were centrifuged and filtered through a 0.22-μm membrane. Water I-class liquid chromatography was used for tandem Xevo tq-s micro triple quadrupole mass spectrometry (Waters, Milford, MA, USA) with an ACQUITY UPLC BEH C18 Column (1.7 μm, 2.1 × 100 mm) (US, Waters). The mobile phase consisted of 0.1% formic acid in water (A) and acetonitrile (B). The injection volume was 5 μL, and the flow rate was set at 0.3 mL/min. The column was maintained at 45 °C. The elution program was as follows: 0–1 min, 2% B; 1–3 min, 2–10% B; 3–10 min, 10–28% B; 10–13 min, 28–60% B; 13–15 min, 60–80% B; 15–16 min, 80–98% B; 16–17 min, 98% B; 17–17.1 min, 98–2% B; 17.1–19 min, 2% B. The conditions of the mass spectrum analysis were as follows: Electron spray ionization (ESI), negative ion mode; the capillary voltage was 2.5 kV; the nozzle voltage was 28.5 V; the gas flow was 10 L/h; the desolvent gas flow was 1100 L/h; the gas temperature was 150 °C; and the desolvent gas temperature was 550 °C. The polyphenols were qualitatively analyzed by the mass spectrum data and the retention time of standards. The limit of detection (LOD) and limit of quantification (LOQ) were calculated by signal-to-noise ratios (S/N) of 3:1 and 10:1, respectively. The compounds were quantified by the standard curves (Appendix A), while compounds without standards were characterized by the mass spectrometry information, retention time, and peak occurrence sequence in [26] (Appendix A); besides, some compounds were quantified by the standard curves of polyphenol compounds with similar structures. Here, 18 standard samples of polyphenols (cynarin, protocatechuic acid, *p*-coumaric acid, caffeic acid, ferulic acid, syringic acid, gallic acid, vanillic acid, *p*-hydroxybenzoic acid, salicylic acid, chlorogenic acid, luteolin, apigenin, phlorizin, and (−)-epicatechin gallate) (>98% purity, Sigma-Aldrich, St Louis, MO, USA) were used for qualitative and quantitative analysis. Additionally, 1-*O*-caffeoylquinic acid and 4-*O*-caffeoylquinic acid were quantified according to the standard curve of chlorogenic acid, and apigenin-7-*O*-glucosidic acid was quantified according to the standard curve of apigenin.

### 4.4. Analysis of Terpenoids

Terpenoids were extracted as described previously by Romas et al. [7]: 5 g of artichoke by-product (on day 0, 2, 5, 10, 20, 30 and 60 of silage) was taken and 125 mL dichloromethane was added. After Soxhlet extraction for 7 h, the extracts were then dried and stored at −20 °C.

The analysis of terpenoids by gas chromatography-mass spectrometry (GC-MS) was carried out as described previously by Romas et al. [7]. Prior to GC-MS detection, trimethylsilanation (TMS) derivatization was performed when necessary (Appendix A). A gas chromatography-mass spectrometer (6890N/5973, Agilent Technologies, 4330 W Chandler Blvd Chandler, AZ, USA) equipped with a DB-5 J and W capillary column (30 m × 0.25 mm × 0.25 μm) (Agilent Technologies, 4330 W Chandler Blvd Chandler, AZ, USA) was used, and the carrier gas was high-purity helium (41 cm/s). The injection volume was 1 μL, the inlet temperature was 270 °C, and the split ratio was 30:1. The oven temperature was programmed as follows: 120 °C for 2 min, then to 250 °C at 4 °C/min, and to 285 °C at 2 °C/min for 15 min. The electron impact mode on the mass spectrometer was set at 70 eV with a mass scan range of 30 to 600 atomic mass units (amu). The ion source temperature was 230 °C. Terpenes and sterols were identified by comparing their mass spectrums with those in the National Institute of Standard and Technology (NIST) mass spectral library and matching their retention indices with the compounds in references [7,48] and standard samples (stigmasterol, β-sitosterol, lupeol, and α-amyrin) (≥95% purity, Sigma-Aldrich, St Louis, MO, USA) when necessary. Qualitative analysis was carried out by n-hexadecane (>99.5 purity, Shanghai Aladdin Bio-Chem Technology Co., LTD, Shanghai, China) and standard samples (Appendix A and Appendix A).

### 4.5. Statistical Analysis

All analyses were performed using three biological replicates (microbial community composition, polyphenols’ and terpenoids’ levels). The results are presented as means ± standard deviations. The test data were processed by one-way ANOVA using the SPSS version 20.0 software package. Multiple comparisons of the data were performed via Tukey’s test, which was used to compute significant differences at *p* < 0.05. Pearson correlation analysis was also used.

## 5. Conclusions

In conclusion, our results suggest that silage is a potential biotechnology for the long-term preservation of bioactive components such as polyphenols and terpenes in artichoke by-products. The results show that the silage of artichoke by-products is characterized by lactic acid bacteria fermentation. The plant’s secondary metabolites resist the invasion of harmful bacteria in the early stages of silage, promoting the growth of lactic acid bacteria, inducing drastic changes for different substances. After that, the lactic acid bacteria promote the production and accumulation of polyphenols and terpenes, and inhibit the growth of harmful bacteria in turn, to ensure the quality of the artichoke by-products. Good silage fermentation is beneficial to enhancing the utilization of artichoke by-products for promoting health and longevity.

## Figures and Tables

**Figure 1 molecules-25-03302-f001:**
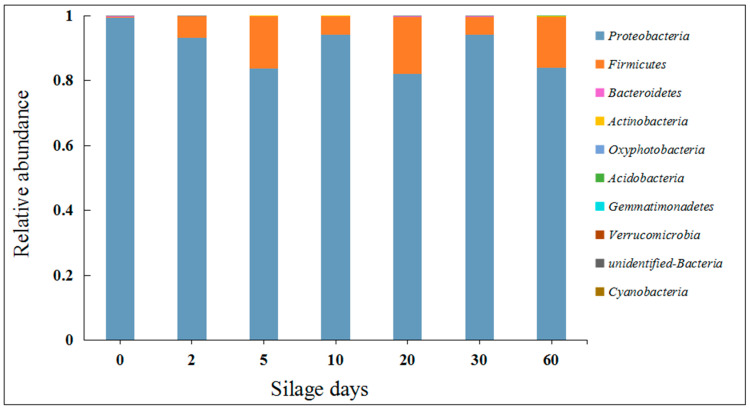
The relative abundances of bacterial communities at the phylum level in silage. The abscissa stands for silage time (on day 0, 2, 5, 10, 20, 30, and 60) and the ordinate stands for the relative abundance of each bacterial community at the phylum level.

**Figure 2 molecules-25-03302-f002:**
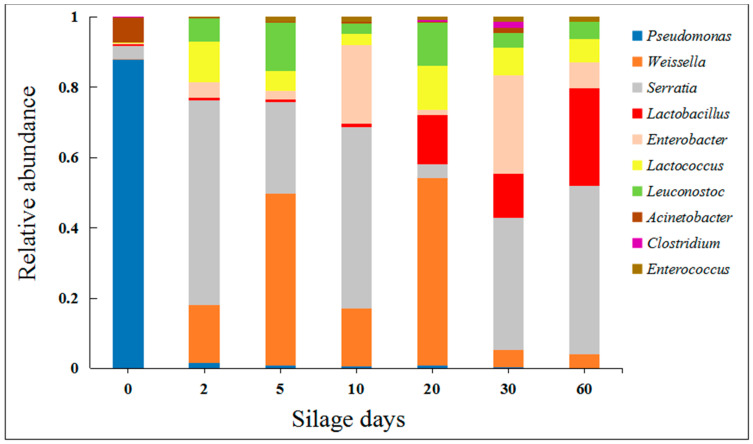
The relative abundance of bacteria communities at the genus level in silage. The abscissa stands for the silage time (on day 0, 2, 5, 10, 20, 30, and 60) and the ordinate stands for the relative abundance of each bacterial community at the genus level.

**Figure 3 molecules-25-03302-f003:**
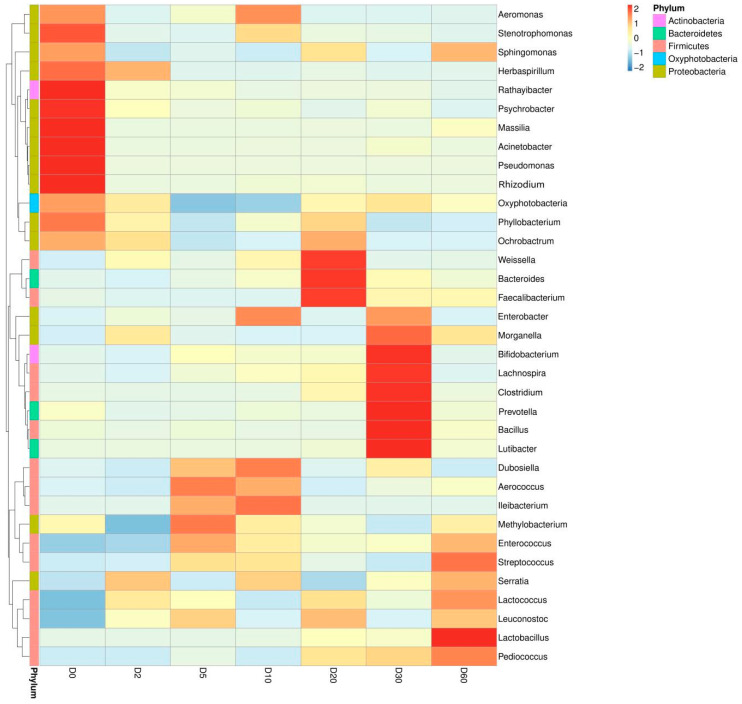
Cluster heat map of bacterial community abundance in silage. The abscissa stands for the silage time (on day 0, 2, 5, 10, 20, 30, and 60) and the ordinate stands for the relative abundance of each bacterial community at the genus level or phylum level.

**Figure 4 molecules-25-03302-f004:**
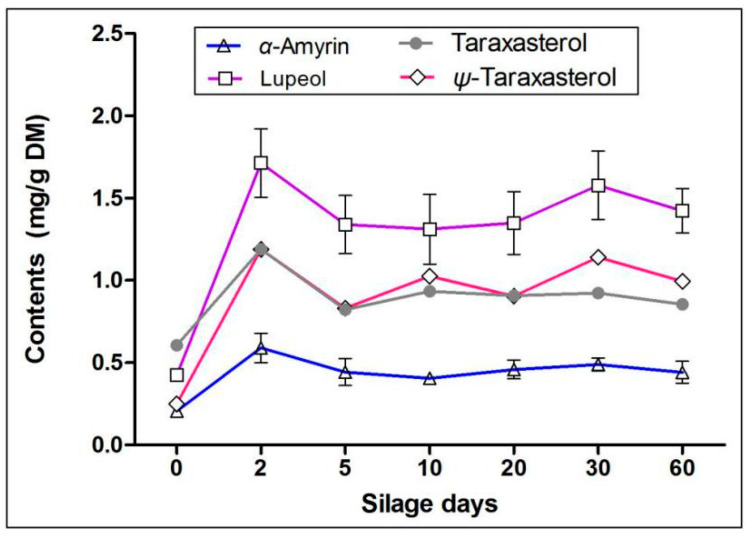
Effects of silage on pentacyclic terpenoids in artichoke by-products.

**Figure 5 molecules-25-03302-f005:**
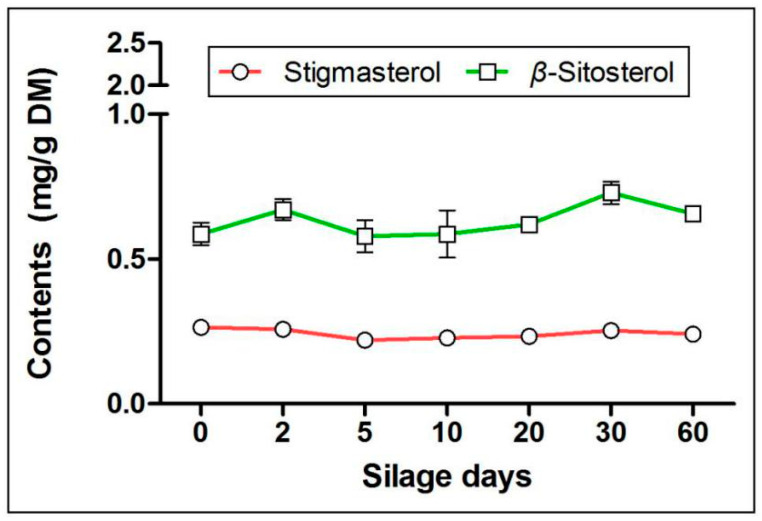
Effects of silage on sterols in artichoke by-products.

**Figure 6 molecules-25-03302-f006:**
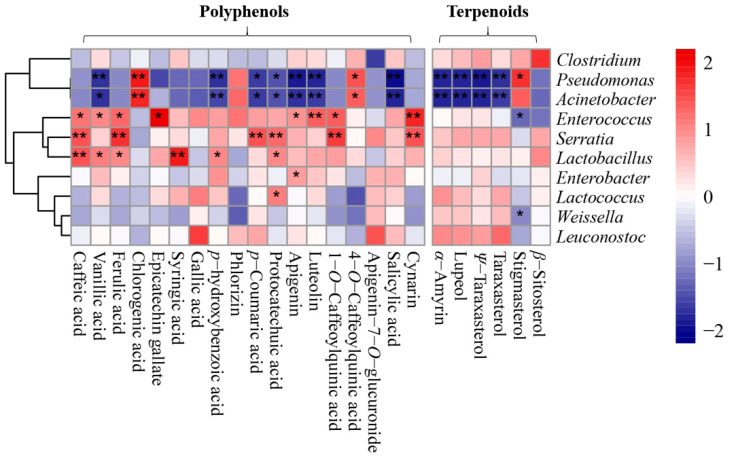
Correlation analysis between the relative abundance of bacterial communities and contents of polyphenols and terpenes in silage. * *p* < 0.01, denoting a significant correlation between the abundance of the bacterial community and levels of the substance with time. ** *p* < 0.05, denoting an extremely significant correlation between the abundance of the bacterial community and levels of the substance with time.

**Figure 7 molecules-25-03302-f007:**
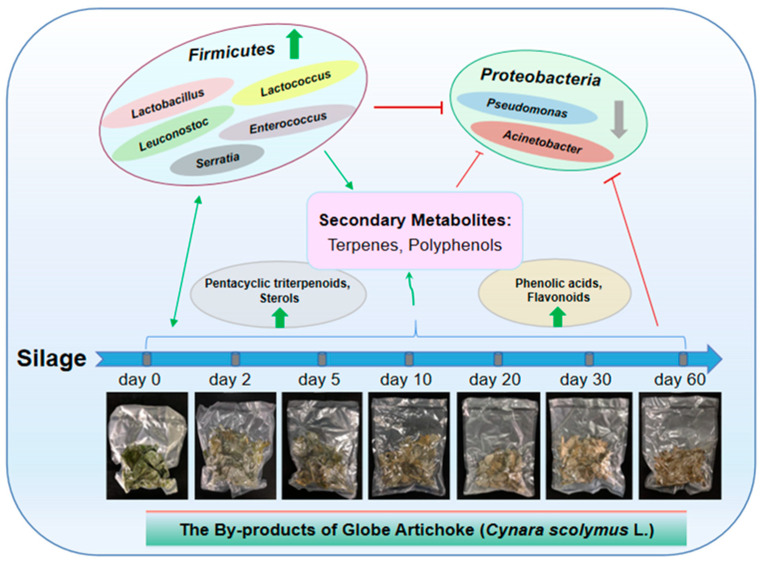
Proposed model depicting the effects of silage on polyphenols and terpenes in globe artichoke (*Cynara scolymus* L.) by-products, involving the changes in the structures of microbial communities. Thick arrow stands for a promotion or inhibition effect, “→” stands for a promotion effect, “*t*” stands for an inhibition effect.

**Table 1 molecules-25-03302-t001:** Effects of silage on phenolic acid composition in artichoke by-products.

Retention Time (min)	Compounds	Silage Days(Day) and Levels of Compounds						
		0 (mg/kg DM)	2 (mg/kg DM)	5 (mg/kg DM)	10 (mg/kg DM)	20 (mg/kg DM)	30 (mg/kg DM)	60 (mg/kg DM)
1.69	Gallic acid	0.41 ± 0.02 ^c^	13.33 ± 2.57 ^b^	34.28 ± 11.37 ^a^	7.82 ± 0.32 ^b,c^	3.71 ± 0.15 ^c^	2.52 ± 0.02 ^c^	4.35 ± 0.12 ^c^
3.01	Protocatechuic acid	1.05 ± 0.06 ^c^	19.36 ± 2.10 ^a^	6.65 ± 1.10 ^b^	6.50 ± 0.16 ^b^	2.11 ± 0.24 ^c^	9.43 ± 0.15 ^b^	18.44 ± 3.35 ^a^
3.12	1-*O*-Caffeoylquinic acid	46.25 ± 1.54 ^c,d^	3.67 ± 0.34 ^e^	13.23 ± 1.82 ^d,e^	12.46 ± 0.50 ^d,e^	50.81 ± 6.12 ^c^	200.45 ± 31.64 ^b^	758.49 ± 36.18 ^a^
3.98	*p*-Hydroxybenzoic acid	1.62 ± 0.12 ^d^	3.80 ± 0.41 ^c^	4.31 ± 0.69 ^c^	5.95 ± 0.58 ^b^	4.62 ± 0.41 ^c^	8.08 ± 0.87 ^a^	7.40 ± 0.69 ^a^
4.64	Caffeic acid	2.61 ± 0.13 ^b^	3.02 ± 0.23 ^b^	3.57 ± 0.33 ^b^	4.75 ± 0.25 ^b^	5.02 ± 0.40 ^b^	15.86 ± 1.23 ^b^	196.24 ± 35.18 ^a^
4.71	Vanillic acid	1.09 ± 0.07 ^f^	2.42 ± 0.21 ^e^	4.11 ± 0.36 ^d^	5.49 ± 0.15 ^c^	5.61 ± 0.61 ^c^	6.72 ± 0.61 ^b^	8.08 ± 0.39 ^a^
5.04	Syringic acid	0.48 ± 0.04 ^d^	0.43 ± 0.04 ^d^	1.18 ± 0.19 ^d^	1.43 ± 0.09 ^d^	3.11 ± 0.32 ^c^	8.86 ± 1.27 ^b^	16.15 ± 1.30 ^a^
5.31	Cynarin	12.41 ± 0.76 ^b^	1.06 ± 0.10 ^d^	0.64 ± 0.09 ^d^	1.64 ± 0.31 ^c,d^	5.78 ± 1.00 ^c^	14.93 ± 0.28 ^b^	112.16 ± 5.01 ^a^
5.81	*p*-Coumaric acid	0.45 ± 0.03 ^f^	1.43 ± 0.12 ^b,c^	1.57 ± 0.14 ^b^	1.16 ± 0.13 ^c,d^	0.94 ± 0.15 ^d,e^	0.78 ± 0.10 ^e^	2.14 ± 0.30 a
6.58	Ferulic acid	1.04 ± 0.09 ^c^	1.52 ± 0.03 ^c^	2.62 ± 0.14 ^b,c^	5.37 ± 0.48 ^b,c^	6.96 ± 0.57 ^b,c^	10.99 ± 1.33 ^b^	125.50 ± 13.57 ^a^
7.97	Salicylic acid	0.05 ± 0.01 ^d^	0.91 ± 0.07 ^c^	1.12 ± 0.21 ^b,c^	1.11 ± 0.03 ^b,c^	1.09 ± 0.17 ^b,c^	1.50 ± 0.06 ^a^	1.32 ± 0.03 ^a,b^
4.51	4-*O*-Caffeoylquinic acid	35.32 ± 1.24 ^a^	1.b62 ± 0.11 ^d^	9.43 ± 0.65 ^b,c^	6.82 ± 0.60 ^c,d^	10.81 ± 1.44 ^b,c^	12.73 ± 1.63 ^b^	38.55 ± 6.00 ^a^
4.33	Chlorogenic acid	6302.87 ± 216.94 ^a^	168.84 ± 14.80 ^b,c^	334.23 ± 44.88 ^b^	132.57 ± 13.09 ^b,c^	238.20 ± 76.87 ^b,c^	46.54 ± 0.18 ^c^	64.11 ± 10.35 ^c^

Values (mean ± SD, *n* = 3) followed by different letters (a to f) in the same row for individual phenolic acids indicate significant differences according to Duncan’s post-hoc test (*p* < 0.05).

**Table 2 molecules-25-03302-t002:** Effects of silage on flavonoid composition in artichoke by-products.

Retention Time (min)	Compounds	Silage Days(Day) and Levels of Compounds						
		0 (mg/kg DM)	2 (mg/kg DM)	5 (mg/kg DM)	10 (mg/kg DM)	20 (mg/kg DM)	30 (mg/kg DM)	60 (mg/kg DM)
6.65	(−)-Epicatechin gallate	0	0.12 ± 0.03 ^d^	0.92 ± 0.21 ^a^	0.60 ± 0.01 ^b^	0.27 ± 0.03 ^c^	0.10 ± 0.08 ^d^	0.85 ± 0.05 ^a^
8.08	Apigenin-7-*O*-glucuronide	58.64 ± 6.82 ^b^	165.22 ± 107.47 ^a^	120.09 ± 3.90 ^a^^,b^	84.61 ± 13.10 ^a,b^	27.74 ± 5.40 ^b^	73.10 ± 1.10 ^a,b^	63.89 ± 6.82 ^b^
9.92	Luteolin	67.67 ± 1.18 ^f^	691.76 ± 83.36 ^e^	1304.59 ± 134.91 ^d^	2416.18 ± 248.61 ^c^	2849.62 ± 138.17 ^b^	3369.26 ± 167.65 ^a^	3627.24 ± 128.40 ^a^
11.21	Apigenin	30.70 ± 3.23 ^f^	219.66 ± 24.39 ^e^	395.00 ± 13.46 ^d^	594.35 ± 34.16 ^b,c^	536.64 ± 29.76 ^c^	639.35 ± 37.46 ^b^	730.59 ± 69.21 ^a^
8.67	Phlorizin	0.57 ± 0.02 ^a^	0.57 ± 0.00 ^a^	0.59 ± 0.00 ^a^	0.57 ± 0.00 ^a^	0.57 ± 0.00 ^a^	0.57 ± 0.01 ^a^	0.57 ± 0.01 ^a^

Values (mean ± SD, *n* = 3) followed by different letters (a to f) in the same row for individual flavonoids indicate significant differences according to Duncan’s post-hoc test (*p* < 0.05).

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
