# Peer review of "Silage Fermentation: A Potential Biological Approach for the Long-Term Preservation and Recycling of Polyphenols and Terpenes in Globe Artichoke (Cynara scolymus L.) By-Products"

_molecules, 2020, doi:10.3390/molecules25143302_

Round 1

Reviewer 1 Report

Manuscript ID: molecules-866351

Title: "Silage Fermentation: a Potential Biological Approach to Long-term Preserving and Recycling Polyphenols and Terpenes in Globe Artichoke (Cynara scolymus L.) By-products"

General comments

The paper analyses the effect of silage on some secondary metabolites in artichoke by-products, such as polyphenols and terpenes.

The basic idea of the manuscript is good, and it could be of practical interest.

However, there are mistakes and some information is missing.

The paper needs correction in terms of style and writing.

Check scientific names: For example: lines 16 and 65- Lactobacillus, line 188- italicize

Homogenize text and nomenclature. For example: “α-amyrin, lupeol, Ψ-taraxasterol, taraxasterol” or “α-Amyrin, Lupeol, Ψ-Taraxasterol, Taraxasterol”; “ml” or “mL”

Separate the units from the numbers

RESULTS

Line 50: Change “Figure1” by “Figure 1”

Figure 1: I can't see the colors of the legend. Axis title is shifted

Figures 2 and 5: Will the genus not be Clostridium instead of Clostridiales?

Figure 2 and 5: Enterobacteriaceae is a family and not a genus

Line 83: Change “table 1” by “Table 1”

Tables: Review the statistic. For example: Is the value of 7.82 equal to 0.42 with those deviations? Same with other values

There are very high deviations in some values. If that is really so, explain why it happens

What does the first column mean in the tables (1, 2, 3, 4, 5,...?

Figure 4: Put the same scale to be able to compare the two graphs

Figure 6: This figure is not commented in the text

DISCUSSION

Check references. In some the number is missing. For example, lines 168-170

MATERIAL AND METHODS

I think that it is necessary to specify details of the fresh plant (state of development, uniformity of samples, harvesting, etc), silage conditions, ... These factors can explain the variations in the results

How many times and how many samples were collected?

Line 197: Change “keshri et al” by “Keshri et al”

Line 266: What treatments?

How many times was the experiment repeated? And the treatments?

REFERENCES

References are not numbered in order of appearance in the text

Check scientific names

Homogenize style

Reviewer 2 Report

The structure of the work is adequate and deserves be published after correction of minor issues.

1) Please consider replace the term flora (or microflora) for microbiota;

2) The sentences in lines 9 and 10 are redundant. Please verify and make in a single sentence. 

3) Lines 50-51: Please consider change “Silage is a process involving complex microbial activitives and nutrient changes, Silage...“ to “Silage is a process involving complex microbial activitives and nutrient changes, this process...“

4) Please consider to make the figure and table titles more informative.

5) Please consider to change the title for table 1 for “Table 1. Effects of silage on phenolic acids composition in artichoke by-products.”
6) Please consider to change the title for table 2 for “Table 2. Effects of silage on flavonoid composition in artichoke by-products.“
7) Please uniformize the “p” symbol (capital letter or small letter?).

8) Line 138: Please correct the term “antiiinflamatory” to “anti-inflammatory”

9) Line 164: “... A total of 18 polyphenols...”

10) Lines 168-172: Please clarify if these studies used the same process of fermentation.

11) Please consider to improve the conclusion section. It is too short.

Reviewer 3 Report

Comments and Suggestions for Authors

The paper entitled “Silage Fermentation: a Potential Biological Approach to Long-term Preserving and Recycling Polyphenols and Terpenes in Globe Artichoke (Cynara scolymus L.) By-products”. In this paper, the potential of silage fermentation for the preservation and recycling of polyphenols and terpenes in artichoke by-products were evaluated.

This work is interesting. The entire manuscript should be corrected.

It is mandatory to correct the manuscript in some points:

#line 33: should be “which is rich in bioactive compounds [2-4].” If you have a string of numbers then shorten the entry. For example [2,3,4] you should write [2-4]. Check and correct all paper.

# Table 1 and 2: name of compounds: “p-Coumaric acid”; p-hydroxybenzoic acid; 4-O-Caffeoylquinic acid; 1-O-Caffeoylquinic acid. O should be italic; p  should be italic.

# please improve the reference.

Reviewer 4 Report

The manuscript by Fan et al describes the potential of silage fermentation for the preservation and recycling of polyphenols and terpenes in artichoke by-products were evaluated. The study provided scientific basis for the efficient utilization of by-products. The manuscript is also generally well organized. Figures and tables are appropriate. Minor English errors are to be fixed. The information generated are of major interest for the readers of Molecules, particularly those with interest in natural products.

Round 2

Reviewer 1 Report

Manuscript ID: molecules-866351-R

Title: "Silage Fermentation: a Potential Biological Approach to Long-term Preserving and Recycling Polyphenols and Terpenes in Globe Artichoke (Cynara scolymus L.) By-products"

General comments

The majority of the comments have been emended in the revised manuscript. Thus, issues on the work, the reproducibility/presentation of results and the explanation of findings are better addressed.

The work is more suitable for publishing